# Calcium Release from Endoplasmic Reticulum Involves Calmodulin-Mediated NADPH Oxidase-Derived Reactive Oxygen Species Production in Endothelial Cells

**DOI:** 10.3390/ijms20071644

**Published:** 2019-04-02

**Authors:** Ryugo Sakurada, Keiichi Odagiri, Akio Hakamata, Chiaki Kamiya, Jiazhang Wei, Hiroshi Watanabe

**Affiliations:** Department of Clinical Pharmacology and Therapeutics, Hamamatsu University School of Medicine, 1-20-1 Handayama, Higashi-ku, Hamamatsu 431-3192, Japan; D16009@hama-med.ac.jp (R.S.); hakamata@hama-med.ac.jp (A.H.); ilka122@hama-med.ac.jp (C.K.); somnent@gmail.com (J.W.); hwat@hama-med.ac.jp (H.W.)

**Keywords:** NADPH oxidase, reactive oxygen species, endothelial cell, endoplasmic reticulum, calcium, calmodulin

## Abstract

Background: Previous studies demonstrated that calcium/calmodulin (Ca^2+^/CaM) activates nicotinamide adenine dinucleotide phosphate oxidases (NOX). In endothelial cells, the elevation of intracellular Ca^2+^ level consists of two components: Ca^2+^ mobilization from the endoplasmic reticulum (ER) and the subsequent store-operated Ca^2+^ entry. However, little is known about which component of Ca^2+^ increase is required to activate NOX in endothelial cells. Here, we investigated the mechanism that regulates NOX-derived reactive oxygen species (ROS) production via a Ca^2+^/CaM-dependent pathway. Methods: We measured ROS production using a fluorescent indicator in endothelial cells and performed phosphorylation assays. Results: Bradykinin (BK) increased NOX-derived cytosolic ROS. When cells were exposed to BK with either a nominal Ca^2+^-free or 1 mM of extracellular Ca^2+^ concentration modified Tyrode’s solution, no difference in BK-induced ROS production was observed; however, chelating of cytosolic Ca^2+^ by BAPTA/AM or the depletion of ER Ca^2+^ contents by thapsigargin eliminated BK-induced ROS production. BK-induced ROS production was inhibited by a CaM inhibitor; however, a Ca^2+^/CaM-dependent protein kinase II (CaMKII) inhibitor did not affect BK-induced ROS production. Furthermore, BK stimulation did not increase phosphorylation of NOX2, NOX4, and NOX5. Conclusions: BK-induced NOX-derived ROS production was mediated via a Ca^2+^/CaM-dependent pathway; however, it was independent from NOX phosphorylation. This was strictly regulated by ER Ca^2+^ contents.

## 1. Introduction

Reactive oxygen species (ROS) have classically been considered a harmful byproduct of oxidant metabolism, as excessive ROS leads to the progression of vascular senescence, atherosclerosis, plaque instability, and endothelial dysfunction in the vascular system [1,2,3]. Now, ROS are recognized as contributors to numerous cellular processes, such as downstream messengers for intracellular signaling [4]. Indeed, ROS play a pivotal role in the regulation of endothelial functions, control of vascular tone, and vascular integrity under physiological conditions. ROS are also involved in vascular cell proliferation, migration, and inflammation in pathophysiological conditions, such as vascular dementia and other cardiovascular diseases [4,5,6,7,8].

The representative enzymatic sources of ROS in endothelial cells are: (1) nitric oxide synthase, (2) the mitochondrial respiratory chain, (3) xanthine dehydrogenase, and (4) nicotinamide adenine dinucleotide phosphate oxidases (NOX). Among several sources of ROS, NOX are predominant in vascular endothelial cells because the NOX family is the main source of endothelial superoxide (O_2_^-^), which is related to endothelial dysfunction and inflammatory activation [4,9]. Furthermore, NOX are a known contributor of redox-sensitive signaling pathways that also lead to endothelial dysfunction [10,11]. Thus, vascular NOX has been identified as a therapeutic target [1,2,12].

Over the past two decades, the regulation of endothelial NOX has been explored and several NOX-activating mechanisms have been identified, including an increase in intracellular calcium ([Ca^2+^]_i_) followed by the subsequent activation of calcium/calmodulin (Ca^2+^/CaM) and Ca^2+^/CaM-dependent protein kinase II (CaMKII) [13,14]. In endothelial cells, store-operated Ca^2+^ entry (SOCE) is a [Ca^2+^]_i_ regulatory mechanism, characterized by the mobilization of Ca^2+^ from the endoplasmic reticulum (ER) and the subsequent Ca^2+^ influx from the extracellular space. Considering that sarcoplasmic reticulum Ca^2+^ release could activate the CaM/CaMKII pathway, as we previously demonstrated in [15], we aimed to determine whether Ca^2+^ mobilization from the ER, but not SOCE, was required to regulate NOX-derived ROS via Ca^2+^/CaM-dependent pathways in porcine aortic endothelial cells (PAECs).

## 2. Results

### 2.1. Bradykinin Increased Cytosolic Ca^2+^ Concentration

Bradykinin (BK) is a widely used pharmacological agent to evoke the SOCE. It causes a biphasic elevation of intracellular Ca^2+^ concentration, which consists of an initial rise caused by Ca^2+^ mobilization from the ER and a subsequent influx of Ca^2+^ from the extracellular space [16,17]. First, we conformed the effect of bradykinin (BK) (1 µM) on cytosolic Ca^2+^ increase in primary cultured porcine aortic endothelial cells (PAECs). For the measurement of cytosolic Ca^2+^, cells were loaded with fura-2/AM (2.5 µM). Figure 1A shows the time courses of changes in the fluorescence ratio (F340/F380) of fura-2 which expressed changes in the intracellular Ca^2+^ concentration. In the presence of 1 mM extracellular Ca^2+^, 1 µM of BK rapidly increased the fluorescence ratio of fura-2 from basal levels of 0.87 ± 0.01 to a maximum of 3.89 ± 0.07 at 90 s (*p* < 0.01), followed by a sustained increase (2.95 ± 0.01 at 360 s, *p* < 0.01 versus baseline). In the absence of extracellular Ca^2+^, BK caused only a small and transient increase (basal levels of 0.59 ± 0.004 to 1.21 ± 0.06 at 90 s, *p* < 0.01). Pretreatment with 1,2-Bis (2 aminophenoxy) ethane-N,N,N’,N-tetraacetic acid tetraacetoxymethyl ester (BAPTA/AM, 100 µM) in Ca^2+^-free modified Tyrode’s solution abolished BK-induced Ca^2+^ responses (basal levels of 0.63 ± 0.004 to 0.64 ± 0.08 at 90 s, *p* = ns) (Figure 1B). These results indicated that 1 µM of BK introduced SOCE, and BAPTA/AM eliminated the effect of BK on ER Ca^2+^ mobilization.

### 2.2. Bradykinin Increased Cytosolic ROS Production

In the following series of experiments, we investigated the effect of BK on cytosolic ROS production in primary cultured PAECs loaded with the ROS indicator 6-carboxy-2′ 7′-dichlorodihydrofluorescein di-(acetate, di-acetoxymethyl ester) (C-DCDHF-DA, 5 µM). Figure 2A shows the time course of changes in the C-DCDHF-DA intensity. In the presence of 1 mM extracellular Ca^2+^, 1 µM of BK rapidly increased the fluorescence intensity of C-DCDHF-DA, which reached the maximum level in 180 s, followed by a sustained increase (F/F_0_ = 1.58 ± 0.05 at 180 s, 1.50 ± 0.05 at 360 s; *p* < 0.05 versus control). This was inhibited by 1 µM of HOE 140, which is an inhibitor of the bradykinin B2 receptor (F/F_0_ = 1.05 ± 0.003 at 180 s, 1.09 ± 0.01 at 360 s; *p* < 0.05 versus BK) (Figure 2B,C). Figure 2D shows representative two-dimensional images of endothelial cells obtained before and at 180 s and 360 s after administration of BK. To confirm that this increase of C-DCDHF-DA signal was caused by increasing cytosolic ROS generation, we evaluated the effect of the free radical scavenger trolox. When cells were pretreated with 100 mM trolox, the increase in the fluorescence intensity of C-DCDHF-DA caused by BK disappeared (Figure 2E,F) (BK: F/F_0_ = 1.58 ± 0.05 at 180 s, 1.50 ± 0.05 at 360 s; trolox: F/F_0_ = 1.07 ± 0.003 at 180 s, 1.09 ± 0.01 at 360 s; *p* < 0.05). This result indicates that BK stimulation increased the cytosolic ROS.

### 2.3. Bradykinin-Induced Cytosolic ROS Production was Caused by NOX Activation

To confirm that BK-induced ROS generation was NOX-dependent, we investigated the effects of NOX inhibitors. Pretreatment with apocynin (3 mM) inhibited the effect of BK (BK: F/F_0_ = 1.36 ± 0.02 at 180 s, 1.33 ± 0.02 at 360 s; apocynin: F/F_0_ = 1.05 ± 0.002 at 180 s, 1.08 ± 0.003 at 360 s; *p* < 0.05) (Figure 3A,B). Furthermore, another NOX specific inhibitor, VAS2870 (50 µM), also significantly suppressed the effects of BK on ROS production (BK: F/F_0_ = 1.45 ± 0.04 at 180 s, 1.55 ± 0.01 at 180 s; VAS2870: F/F_0_ = 1.11 ± 0.002 at 180 s F/F_0_ = 1.25 ± 0.01 at 360 s; *p* < 0.05) (Figure 3C,D). These results indicated that BK-induced ROS production was caused by NOX activation.

### 2.4. Endoplasmic Reticulum Ca^2+^ and BK-Induced NOX-Derived ROS Production

Next, we investigated the involvement of intra- or extra-cellular Ca^2+^ in the effects of BK. Figure 4A shows the time course of changes in the effects of BK on NOX-derived ROS production in the absence or presence of extracellular Ca^2+^. When cells were exposed to BK with a nominal Ca^2+^-free ([Ca^2+^]_o_ = 0 mM with 1 mM EGTA) or [Ca^2+^]_o_ = 1 mM modified Tyrode’s solution, no difference in the effects of BK on NOX-derived ROS production was observed (0 mM Ca^2+^ with 1 mM EGTA: F/F_0_ = 1.79 ± 0.04 in 180 s, 1.88 ± 0.03 at 360 s; [Ca^2+^]_o_ = 1 mM: F/F0 = 1.77 ± 0.04 in 180 s, 1.83 ± 0.04 at 360 s; not significant) (Figure 4B). When BAPTA/AM (100 µM) was added to modified Tyrode’s solution ([Ca^2+^]_o_ = 0 mM) for chelating of intracellular Ca^2+^, the effect of BK on NOX-derived ROS production was completely attenuated (0 mM Ca^2+^ with 1 mM EGTA: F/F_0_ = 2.02 ± 0.05 at 180 s, 2.03 ± 0.05 at 360 s; BAPTA/AM: F/F_0_ = 1.05 ± 0.002 at 180 s, 1.10 ± 0.003 at 300 s; p < 0.05) (Figure 4C,D). Furthermore, pretreatment with thapsigargin (TG, 1 µM) in nominal Ca^2+^-free modified Tyrode’s solution for depletion of ER Ca^2+^ content also completely abolished the effects of BK on NOX-derived ROS production (Figure 3E,F) (0 mM Ca^2+^ with 1 mM EGTA: F/F_0_ = 1.79 ± 0.04 at 180 s, 1.88 ± 0.03 at 360 s; TG: F/F_0_ = 1.09 ± 0.01 at 180 s, 1.16 ± 0.01 at 360 s, *p* < 0.05). Taken together, these results indicated that ER Ca^2+^ contents and local Ca^2+^ mobilization from the ER were required to activate NOX, whereas SOCE was not involved in this phenomenon.

### 2.5. The Ca^2+^/CaM Pathway Regulated BK-Induced NOX-Derived ROS Production

CaM, one of the most ubiquitous Ca^2+^-sensing proteins, regulates various cellular functions via activation of downstream kinases, such as CaMKII. To determine whether Ca^2+^/CaM and the CaMKII pathway were involved in BK-induced NOX-derived ROS production, we examined the effects of W-7 (a CaM inhibitor) and KN-93 (a CaMKII inhibitor) on BK-induced NOX-derived ROS production. With W-7 pretreatment (100 mM), the effect of BK on NOX-derived ROS production was completely abolished (BK: F/F_0_ = 1.58 ± 0.03 at 180 s, 1.71 ± 0.03 at 360 s; W-7: F/F_0_ = 1.10 ± 0.01 at 180 s, 1.21 ± 0.01 at 360 s; *p* < 0.05) (Figure 5A,B). However, KN-93 (100 µM) did not show an inhibitory effect on BK-induced NOX-derived ROS production (BK: F/F_0_ = 1.55 ± 0.03 at 120 s, 1.64 ± 0.04 at 180 s; KN-93: F/F_0_ = 1.55 ± 0.03 at 120 s, 1.69 ± 0.04 at 360 s, not significant) (Figure 5C,D). These results indicated that BK-induced NOX-derived ROS production was mediated via a Ca^2+^/CaM-dependent, CaMKII-independent pathway.

### 2.6. The Ca^2+^/CaM Pathway Regulated BK-Induced NOX-Derived ROS Production

We finally examined whether other Ca^2+^/CaM-dependent protein kinases, such as CaMKI and CaMKIV, could phosphorylate NOX family members. We performed a phosphorylation assay using a Phos-tag™ SDS-PAGE Western blotting method. As shown in Figure 6, BK did not phosphorylate glycosylated NOX2, or unglycosylated NOX2, NOX4, or NOX5. These results show that phosphorylation enzymes were not associated with CaM-mediated NOX activation.

## 3. Discussion

Spatial and temporal increases in [Ca^2+^]_i_ participate in several types of cellular signaling. Changes in [Ca^2+^]_i_ regulate the enzymatic activities of several sources of ROS, such as endothelial nitric oxide synthase, mitochondria, and NOX in vascular smooth muscle cells and neutrophils [18,19]. Calcium influx from the extracellular space is a major source of elevated [Ca^2+^]_i_, and SOCE plays a central role of this increase in [Ca^2+^]_i_ in non-excitable cells. Indeed, several reports have demonstrated that an increase in [Ca^2+^]_i_ via SOCE activates NOX and promotes O_2_^-^ generation [20,21,22]. In the current study, we focused on the intracellular Ca^2+^ via Ca^2+^ release from the ER and investigated the role of [Ca^2+^]_i_ elevation on the activation of NOX in PAECs. The major findings demonstrated that an increase in intracellular Ca^2+^ via ER Ca^2+^ mobilization (but not SOCE), which was introduced by BK stimuli, could activate NOX and it involved ROS generation. We also elucidated that the mechanism underlying this BK-induced NOX activation was through a Ca^2+^/CaM-dependent and phosphorylation-independent process. Our findings clarify the interplay between ER Ca^2+^ mobilization and regulation of NOX-derived ROS production in PAECs.

Bradykinin is an agonist of bradykinin receptor B2, a G-protein coupled receptor (GPCR), expressed on the surface of endothelial cells. Bradykinin stimuli activates the GPCR, resulting in a biphasic increase in [Ca^2+^]_i_ (so called SOCE) via the phospholipase-C-inositol 1,4,5-triphosphate signaling pathway [23,24]. Indeed, we and our colleagues revealed that BK (10 nM) could increase [Ca^2+^]_i_ via ER Ca^2+^ release, followed by SOC-mediated extracellular Ca^2+^ influx in PAECs [16,25,26]. In addition, Ana Rita Pinheiro et al. also reported that BK (0.001–100 μM) could increase [Ca^2+^]_i_ in a dose-dependent manner [27]. Furthermore, our colleagues also revealed that 1µ M of BK activated endothelial nitric oxide synthase, which requires Ca^2+^ entry for sustained activation [28]. On the basis of these findings, we confirmed that 1 µM of BK evoked SOCE in PAECs and used 1 µM of BK to introduce the increase of [Ca^2+^]_i_. Agonists of GPCR, including angiotensin II and endothelin-1, were shown to increase endothelial NOX activity [29,30]. Although previous reports demonstrated that BK activated NOX, followed by increased ROS formation in brain astrocytes [31,32], the effect of BK on endothelial NOX and ROS generation had not been fully elucidated. Here, we used a fluorescent dye technique to monitor endothelial ROS generation, which clearly demonstrated that BK stimuli increased ROS production via endothelial NOX activation. In this study, we used two types of pharmacological NOX inhibitors because they had different NOX inhibitory properties. Apocinin, which is historically the most frequently used NOX inhibitor, has ROS-scavenging properties [33]. Whereas, VAS2870 is a NOX-specific, but not isoform-selective, inhibitor that has no ROS-scavenging property [34]. From our results, both VAS2870 and apocinin significantly suppressed the effects of BK on endothelial ROS production, strongly indicating that BK increased endothelial cytosolic ROS formation via the activation of NOX.

Intracellular Ca^2+^ plays an essential role in initiating various intracellular signaling cascades. Various plasma membrane channels, such as L-type voltage-dependent Ca^2+^ channels and transient receptor canonical channels implicate Ca^2+^ influx from extracellular spaces [35,36]. In endothelial cells, store-operated Ca^2+^ channels form a major pathway for initiation of intracellular Ca^2+^ signaling. Although several downstream pathways of Ca^2+^ signaling, including CaM, CaMKII, and protein kinase C were identified as NOX activators [13,14,32,37,38,39], evidence has not established which components of intracellular Ca^2+^ increase, ER Ca^2+^ release, or the subsequent Ca^2+^ influx from extracellular spaces are required to regulate NOX-derived ROS production in endothelial cells. We showed that extracellular Ca^2+^ removal (in the presence of 1 mM EGTA) to eliminate Ca^2+^ influx from extracellular spaces could not impact BK-induced ROS production, while quenching of intracellular Ca^2+^ by BAPTA/AM abolished BK-induced ROS production. Furthermore, depletion of ER Ca^2+^ contents by TG also abolished BK-induced ROS production. These results strongly suggested that NOX activation required the elevation of [Ca^2+^]_i_ evoked by Ca^2+^ release from the ER.

The CaM/CaMKII pathway is known as a ubiquitous downstream pathway of Ca^2+^ signaling in the vascular system [40,41], and both contribute to NOX regulation [13,14,38]. We demonstrated that W-7, an inhibitor of CaM, abolished the effect of BK on ROS generation, whereas KN-93, an inhibitor of CaMKII, had no effect. Furthermore, an in vivo phosphorylation assay revealed that BK stimuli did not cause phosphorylation of NOX2, 4, or 5, which are expressed in endothelial cells. These results suggested that CaM regulated NOX activity by itself, independent of Ca^2+^/CaM-dependent phosphoenzymes. Although protein phosphorylation is an essential step for activation of NOX family members, [42,43] phosphorylation was not necessarily required to activate NOX5 [38,44]. Instead, NOX5 was activated by Ca^2+^, while some protein kinases, such as protein kinase C and CaMKII, modulated its enzymatic activity [14,38,39,44]. Importantly, NOX5 has a CaM-binding domain at its C-terminus. When CaM bound to the CaM-binding domain of NOX5, the C-terminus conformation was changed to increase Ca^2+^ sensitivity of N-terminus-regulated enzymatic activity [38]. This previous evidence is consistent with our finding that NOX is activated by Ca^2+^/CaM without phosphorylation and it suggested that NOX5 might be involved in Ca^2+^/CaM-mediated ROS production in our experiments.

## 4. Materials and Methods

### 4.1. Cell Culture

Porcine aortic endothelial cells were harvested from porcine aortas (collected from a local slaughterhouse), then isolated and cultured as previously described in [45] by gently scraping the intima of the descending part of the porcine aorta. After centrifugation at 1400× *g* for 15 min in Medium 199 (M-199), the pellet of PAECs was purified from this suspension, resuspended in M-199 medium with Earle’s salts, and supplemented with 100 IU/mL penicillin G, 100 µg/mL streptomycin, and 20% newborn calf serum. Cells were then seeded onto culture dishes or silicone dishes fixed on glass coverslips and incubated at 37 °C in 5% CO_2_. This study conformed to the Guide for the Care and Use of Laboratory Animals published by the US National Institutes of Health (NIH publication, 8th edition, 2011). All experiments were performed in accordance with the regulations of the Animal Research Committee of the Hamamatsu University School of Medicine.

### 4.2. Measurement of Endothelial Ca^2+^ Concentration and ROS Production

Intracellular Ca^2+^ concentration was measured using fura-2/AM, a fluorescent Ca^2+^ indicator. Porcine aortic endothelial cells were incubated in modified Tyrode’s solution composed of (mM) 150 NaCl, 2.7 KCl, 1.2 KH_2_PO_4_, 1.2 MgSO_4_, 1.0 CaCl_2_, and 10*N*-2-hydroxyethylpiperazine-*N*-2-ethanesulfonic acid (pH 7.4 at 25 °C) with fura-2/AM (2.5 μM) for 45 min. Intracellular ROS was measured using the fluorescent indicator C-DCDHF-DA. Cells were incubated in modified Tyrode’s solution with C-DCDHF-DA (5 μM) for 20 min.

Before measurements of Ca^2+^ and ROS, cells were subsequently washed three times with modified Tyrode’s solution to remove the dye. The absorption shift of fura-2 that occurs upon binding to intracellular Ca^2+^ is determined by scanning the excitation spectrum between 340 and 380 nm while monitoring the emission at 510 nm. C-DCDHF-DA was excited at 490 nm with a xenon lamp, and emission signals were collected through a 510-nm long-pass filter. Fluorescent images of fura-2/AM and C-DCDHF-DA were acquired and quantified every 30 s from individual cells with a fluorescence analyzer (Aqua-cosmos, Hamamatsu Photonics K.K, Hamamatsu, Japan) using an ultra-high sensitivity camera. Changes in the fluorescence ratio (F340/F380) of fura-2 were used to express changes in the intracellular Ca^2+^ concentration. Changes in C-DCDHF-DA fluorescence intensities (F) in each experiment were normalized to the fluorescence intensity recorded at the time of BK administration for stimulating ER Ca^2+^ release (F_0_). All experiments were performed at 25 °C.

### 4.3. Phos-tag™ SDS-PAGE Western Blotting

After addition of BK, cells were washed with ice-cold Ca^2+^-free phosphate-buffered saline, treated with lysis buffer containing protease inhibitors and phosphatase inhibitors (Ez RIPA Lysis Kit™, Tokyo, Japan) for 15 min at 4 °C, then scraped and harvested. Samples containing equal amounts (10 µg) of total cellular protein were loaded and separated on 10% SDS-polyacrylamide gel with 50 µM Phos-tag (SuperSep Phos-tag™, Tokyo, Japan). Electrophoresis was carried out at 20 mA/gel for 80 min. The gel was then transferred onto a polyvinylidene difluoride membrane at 153 mA/membrane for 30 min. Blocking was carried out at 4 °C overnight using a blocking agent (EzBlock Chemi™; ATTO, Tokyo, Japan). The membrane was stained with primary antibodies against NOX2 (1:2000 dilution), NOX4 (1:2000 dilution), and NOX5 (1:2000 dilution) for 1 h at room temperature. Phosphorylated or unphosphorylated proteins bound to each primary antibody were probed with a horseradish peroxidase-conjugated secondary antibody for 1 h at room temperature (1:5000 dilution). Protein bands were visualized with a ChemiDoc Touch system (Bio-Rad Laboratories, Hercules, CA, USA).

### 4.4. Chemicals

M-199, 100 IU/mL penicillin G and 100 µg/mL streptomycin were purchased from Gibco. Newborn calf serum was purchased from Life Technology. BK, thapsigargin, VAS2870, and HOE-140 were purchased from Sigma-Aldrich. KN-93, W-7 hydrochloride, Trolox, and Phos-tag were purchased from Wako. Fura-2/AM and BAPTA/AM were purchased from Dojindo. C-DCDHF-DA was purchased from Invitrogen. EzBlock Chemi™ and Ez RIPA Lysis Kit™ were purchased from ATTO. Anti-NOX2, 3, and 5 antibodies were purchased from Proteintech. Apocynin and horseradish peroxidase-conjugated secondary antibody were purchased from Santa Cruz Biotechnology.

### 4.5. Statistics Analysis

Data are expressed as the mean ± SEM, and the number of cells or experiments were shown as *n*. Statistical analysis was performed by two-way factorial ANOVA followed by the Bonferroni test at each measurement point. A level of *p* < 0.05 was accepted as statistically significant. Computations were performed using SPSS Statistics version 24.0 (IBM Corporation, Armonk, NY).

## 5. Conclusions

The highlights of our study are: (1) BK increases Ca^2+^ mobilization from ER, and this increase of local [Ca^2+^]_i_ activates the Ca^2+^/CaM pathway; (2) the Ca^2+^/CaM pathway directly activates NOX, leading to increased cytosolic ROS production; (3) Ca^2+^ mobilization from ER, but not store-operated Ca^2+^ entry, regulates CaM-mediated NOX-derived ROS production; and (4) we could not detect which NOX subtypes were activated by BK stimulation. We concluded that CaM could directly activate endothelial NOX independently from phosphorylation, and it accelerated ROS production in PAECs. This effect of CaM was strictly controlled by an elevation of [Ca^2+^]_i_ evoked by ER Ca^2+^ mobilization. Further studies will be needed to elucidate which of the NOXs are engaged in BK-induced ROS production.

## Figures and Tables

**Figure 1 ijms-20-01644-f001:**
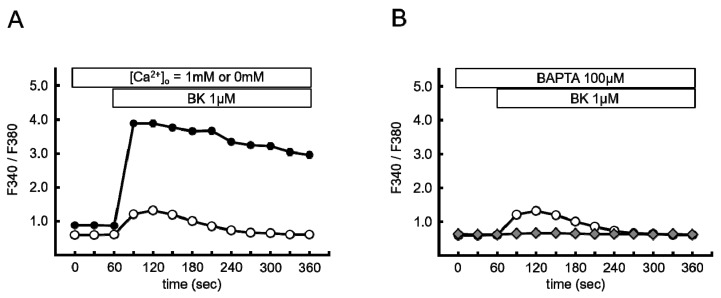
Bradykinin (BK) increased cytosolic Ca^2+^ concentration. (**A**) Time course of changes in the fluorescence ratio (F340/F380) of fura-2 in Ca^2+^-free (open circle; *n* = 255) and 1 mM Ca^2+^ (closed circle; *n* = 210) modified Tyrode’s solution. (**B**) The same experiment as (**A**) conducted in Ca^2+^-free modified Tyrode’s solution with (open circle; *n* = 255) or without 1,2-Bis (2 aminophenoxy) ethane-N,N,N’,N-tetraacetic acid tetraacetoxymethyl ester (BAPTA) (closed diamond; *n* = 203). Data are expressed as the mean ± standard error of the mean (SEM) of three independent experiments in separate cell culture wells.

**Figure 2 ijms-20-01644-f002:**
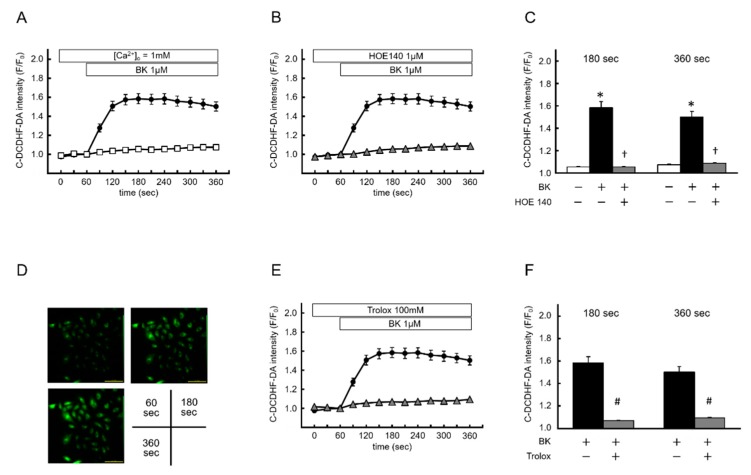
Bradykinin induces intracellular reactive oxygen species. (**A**,**B**) Time course of changes in 6-carboxy-2′ 7′-dichlorodihydrofluorescein di-(acetate, di-acetoxymethyl ester) (C-DCDHF-DA) intensity. Porcine aortic endothelial cells were incubated with modified Tyrode’s solution (control, [Ca^2+^]_o_ = 1mM, open square; *n* = 53), and then BK was applied (closed circle; *n* = 111). HOE 140 (open triangle; *n* = 107) significantly suppressed the BK-induced increase of C-DCDHF-DA intensity. (**C**) Summarized data of C-DCDHF-DA intensity at 180 s and 360 s with or without HOE 140. (**D**) Two-dimensional images of C-DCDHF-DA 60 s (before BK application) (upper left), 180 s (upper right), and 360 s 6 (lower left). (**E**,**F**) The same experiments as (**A**) conducted in the presence (open triangle; *n* = 185) or absence (closed circle; *n* = 111) of trolox. Increased C-DCDHF-DA intensity reflects cytosolic reactive oxygen species production. Data are expressed as the mean ± SEM of three independent experiments in separate cell culture wells. * *p* < 0.01 versus control, ^†^
*p* < 0.01 versus BK, and ^#^
*p* < 0.05 versus BK. Two-factor factorial ANOVA was used for analysis of difference.

**Figure 3 ijms-20-01644-f003:**
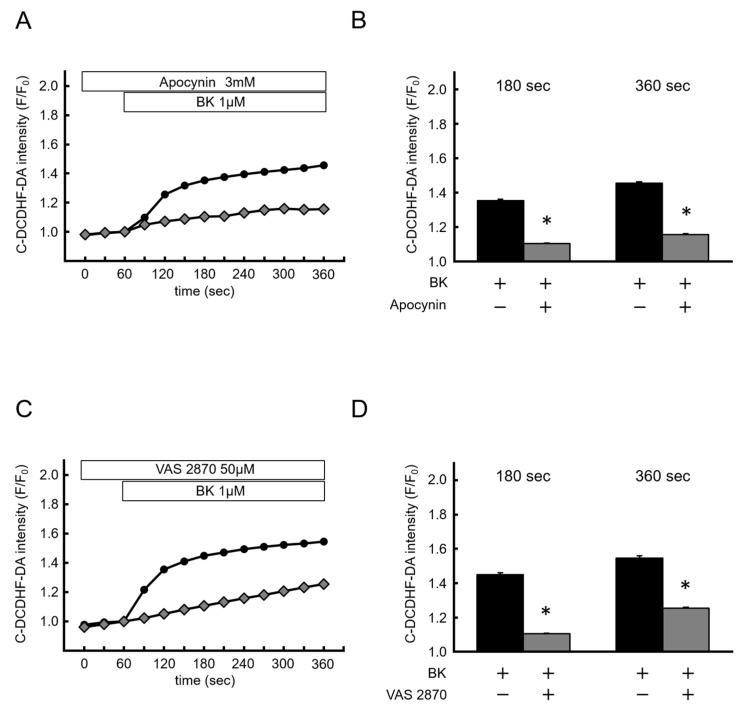
The source of bradykinin-induced reactive oxygen species is NADPH oxidases. (**A**,**B**) Time course of changes in C-DCDHF-DA intensity, and summarized data of C-DCDHF-DA intensity at 180 s and 360 s in the presence (open diamond; *n* = 336) or absence (closed circle; *n* = 300) of VAS2870. (**C**,**D**) The same experiment as (**A**) conducted in the presence (open diamond; *n* = 308) or absence (closed circle; *n* = 201) of apocinin. Data are expressed as the mean ± SEM of 3–5 independent experiments in separate cell culture wells. * *p* < 0.05 versus BK. Two-factor factorial ANOVA was used for analysis of difference.

**Figure 4 ijms-20-01644-f004:**
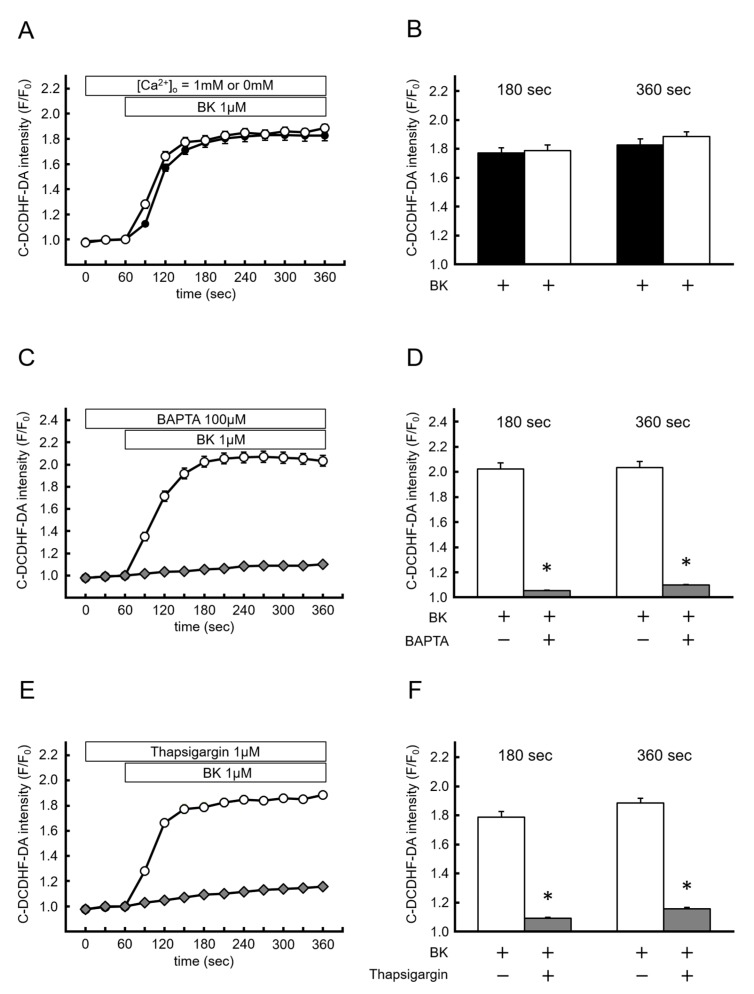
Ca^2+^ release from endoplasmic reticulum is required for bradykinin-induced reactive oxygen species production. (**A**,**B**) Time course of changes in C-DCDHF-DA intensity, and summarized data of C-DCDHF-DA intensity at 180 s and 360 s in Ca^2+^-free (open circle; *n* = 181) and 1 mM Ca^2+^ (closed circle; *n* = 263) modified Tyrode’s solution. (**C**,**D**) The same experiment as (**A**) conducted in Ca^2+^-free modified Tyrode’s solution with (open diamond; *n* = 174) or without BAPTA (open circle; n = 146). (**E,F**) The same experiment as (**A**) conducted in Ca^2+^-free modified Tyrode’s solution with (open diamond; *n* = 123) or without thapsigargin (open circle; *n* = 181). Data are expressed as the mean ± SEM of 3–5 independent experiments in separate cell culture wells. * *p* < 0.01 versus BK. Two-factor factorial ANOVA was used for analysis of difference.

**Figure 5 ijms-20-01644-f005:**
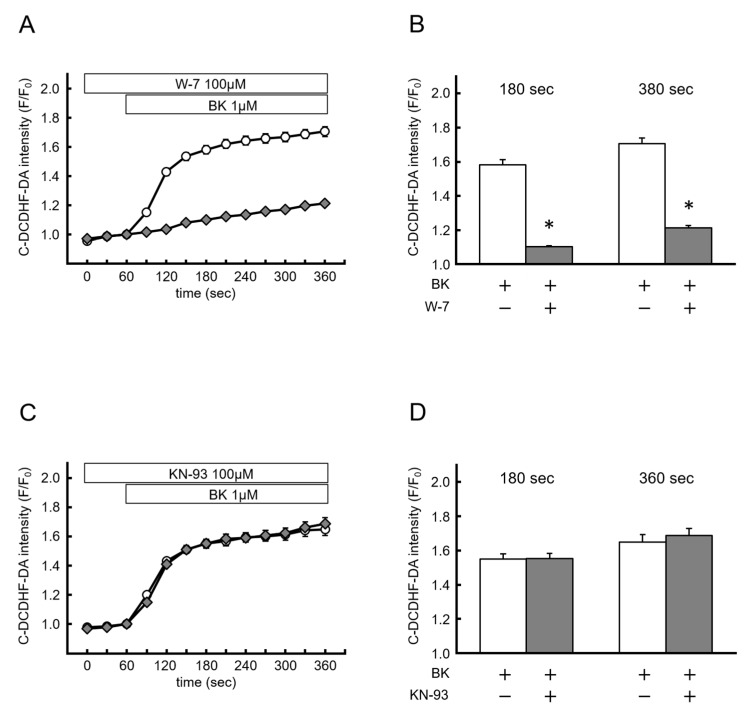
NADPH oxidase-derived reactive oxygen species production by bradykinin is mediated via a Ca^2+^/Calmodulin-dependent pathway. (**A**,**B**) Time course of changes in C-DCDHF-DA intensity, and summarized data of C-DCDHF-DA intensity at 180 s and 360 s after BK application in the presence (open diamond; *n* = 93) or absence (open circle; *n* = 136) of W-7. (**C**,**D**) The same experiments as (**A**) conducted in the presence (open diamond; *n* = 63) or absence (open circle; *n* = 136) of KN-93. Data are expressed as the mean ± SEM of two independent experiments in separate cell culture wells. * *p* < 0.01 versus BK. Two-factor factorial ANOVA was used for analysis of difference.

**Figure 6 ijms-20-01644-f006:**
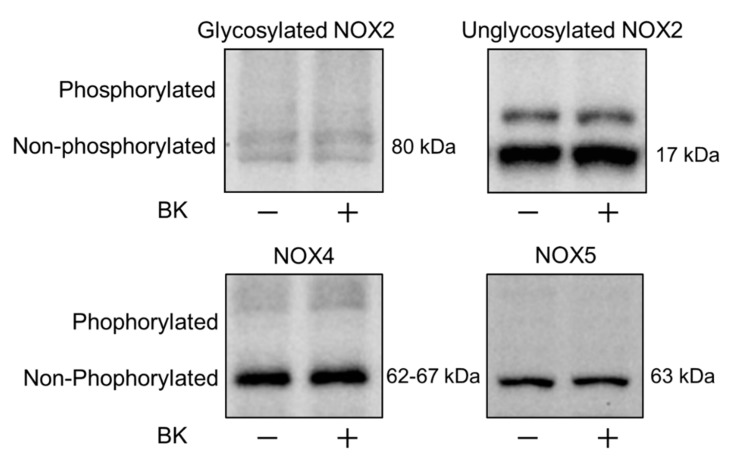
Bradykinin stimuli did not mediate NADPH oxidase 2, 4, or 5. Representative Phos-tag™ SDS-PAGE Western blots of glycosylated NOX2 and unglycosylated NOX2, NOX4, and NOX5. There was no phosphorylation of the core protein of each NOX by BK stimulation.

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
