# Peer review of "Calcium Release from Endoplasmic Reticulum Involves Calmodulin-Mediated NADPH Oxidase-Derived Reactive Oxygen Species Production in Endothelial Cells"

_ijms, 2019, doi:10.3390/ijms20071644_

Reviewer 1 Report

This article was revised appropriately.

I recommend accept

Reviewer 2 Report

The authors have revised the manuscript according to the reviewer's comments.

This manuscript is a resubmission of an earlier submission. The following is a list of the peer review reports and author responses from that submission.

Round  1

Reviewer 1 Report

The subject of the article is interesting, once the effect of Bradykinin‑induced NOX‑derived ROS production was analysed, and the authors observed that are mediated via a Ca2+/CaM‑dependent pathway and independent from NOX phosphorylation.

General comments:

1-     In the legends of figures, the static test used needs to be added.

2-     In the figure 1C and 1F the authors used two‑way ANOVA followed by Bonferroni test? The same for figure 2B, D, 3B, 3D, 3F, 4B and 4D?  The number of parameters are only 2?

3-     I the figure 1 A, B, E, pleased explain the -60sec?

4-     The references to the figures should be eliminated from the discussion.

5-     The conclusions must be improved and increased.

6-     Extensive editing of English language and style required.

Author Response

 Response to Reviewer 1

Comments and Suggestions: The subject of the article is interesting, once the effect of Bradykinin-induced NOX-derived ROS production was analysed, and the authors observed that are mediated via a Ca2+/CaM-dependent pathway and independent from NOX phosphorylation.

Answer: We thank you very much for the reviewer’s motivated and positive comments about our study.

We have tried to respond the reviewer’s comments and resolve several issues. We think our manuscript has greatly improved by the reviewer’s suggestion.

Comment 1: In the legends of figures, the static test used needs to be added.

Response: We appreciate the Reviewer’s comment on this point. As described below and 4.4. Statistics analysis section (page. 15, lines 288-289), we used two-factor factorial ANOVA for the comparison of the difference.
We have added the following text in the figure legends of figure 1, 2, 3 and 4:

 “Two-factor factorial ANOVA was used for analysis of difference.”

(Added to figure legend of figure 1. page 3, lines 97-98.)

(Added to figure legend of figure 2. page 4, line 116.)

(Added to figure legend of figure 3. page 6, line 148.)

(Added to figure legend of figure 4. page 8, lines 170-171.)

 Comment 2: In the figure 1C and 1F the authors used two-way ANOVA followed by Bonferroni test? The same for figure 2B, D, 3B, 3D, 3F, 4B and 4D? The number of parameters are only 2?

Response: We are sorry for incomplete statistical information. We used two-factor factorial ANOVA for the comparison of the difference, and we analyzed the difference of the C-DCDHF-DA intensity at each measurement point (figure 1A, 1B and 1E). We also showed the representative measurement values of C-DCDHF-DA intensity at 120 sec (as a maximum level) and 300 sec (as a plateau level) after BK application in figure 1C and 1F. The same analysis was used for figure 2B, D, 3B, 3D, 3F, 4B and 4D.

We have changed the 4.4. Statistics analysis section (page 15, lines 299-300) as followed:

“Statistical analysis was performed by two-way factorial ANOVA followed by Bonferroni test at each measurement point.

Comment 3: In the figure 1 A, B, E, pleased explain the -60sec?

Response: We are sorry for the inadequate information. We set the bradykinin appreciation time as a start time (0 second) of the experiment. In this revised manuscript, we correct the figure 1A, 1B, 1E, 2A, 2C, 3A, 3C, 3E, 4A and 4C. We set the measurement starting point as a zero second.

We have replaced the Figure 1A, 1B, 1E, 2A, 2C, 3A, 3C, 3E, 4A and 4C (page 3, 4, 6 and 7)

In addition, we have correct the each measurement time in results section (including figure legends):
(120 sec to 180 sec, and 300 sec to 360 sec) 

Comment 4: The references to the figures should be eliminated from the discussion.

Response: We appreciate the Reviewer’s comment on this point.

In accordance with Reviewer’s comment, we removed the references to the figures from discussion section (page 8, lines 212, 213, 230-232, and 236-237).

Comment 5: The conclusions must be improved and increased.

Response: We appreciate the Reviewer’s comment on this point. We re-wrote the conclusions.

We have added the following text in Conclusion section (page 11, lines 305-310):

The highlights of our study are (1) BK increases Ca2+ mobilization from ER, and this increase of local [Ca2+]i activates the Ca2+/CaM pathway; (2) the Ca2+/CaM pathway directly activates NOX, leading to increased cytosolic ROS production; (3) Ca2+ mobilization from ER, but not store-operated Ca2+ entry, regulates CaM-mediated NOX-derived ROS production; and (4) we could not detect which NOX subtypes were activated by BK stimulation.”

page 11, lines 312-313:

Further studies will be needed to elucidate which one of the NOXs or all of them are engaged in BK-induced ROS production.

We appreciate your comment because only after reading your comment did we note that in the final version of this manuscript we made a typing error when we converted Ca2+ to Ca2+ The error has been corrected in the revised manuscript. Please, see the correction made in the Conclusion section, page 10, line 289. 

Comment 6: Extensive editing of English language and style required

Response: We appreciate the Reviewer’s comment on this point. According to your suggestion, we have our manuscript checked by a professional English editing service.

Reviewer 2 Report

The authors examined bradykinin (BK)–induced NOX activation and ROS generation in endothelial cells and the involvement of calcium and the calcium source responsible for using pharmacological interventions. The results suggest that BK induces NOX activation and subsequent ROS generation that depends on intracellular calcium release but not on extracellular calcium influx. The study further demonstrated that CaM, but not calcium/CaM kinase II, is critically involved.  The manuscript is generally well written. However,  the authors should perform control experiments to show BK-induced calcium responses and additional experiments to clarify the NOX subtypes are engaged in mediating BK-induced ROS generation.

Major points:

1. The study need to demonstrate experimentally that BK at the concentration used can induce internal calcium release and SOC-mediated extracellular calcium influx.  Such information is important for the interpretation of the experiments shown in the study.

2. The study showed the role of NOX in BK-induced ROS generation using genic NOX blockers, apocynin or VAS. The authors also demonstrated by western blotting expression of several NOX subtypes, NOX2, NOX4 and NOX5 in endothelial cells.  It is highly desirable or interesting to explore which one of NOXs or all of them are engaged in BK-induced ROS production.

 Author Response

Response to Reviewer 2
Comments and Suggestions: The authors examined bradykinin (BK)–induced NOX activation and ROS generation in endothelial cells and the involvement of calcium and the calcium source responsible for using pharmacological interventions. The results suggest that BK induces NOX activation and subsequent ROS generation that depends on intracellular calcium release but not on extracellular calcium influx. The study further demonstrated that CaM, but not calcium/CaM kinase II, is critically involved. The manuscript is generally well written. However, the authors should perform control experiments to show BK-induced calcium responses and additional experiments to clarify the NOX subtypes are engaged in mediating BK-induced ROS generation

Answer: We thank very much for the reviewer for the important and constructive criticism of our manuscript. We have tried to respond the reviewer’s comments. We think our manuscript has greatly improved by the reviewer’s suggestion.

Comment 1: The study need to demonstrate experimentally that BK at the concentration used can induce internal calcium release and SOC-mediated extracellular calcium influx. Such information is important for the interpretation of the experiments shown in the study

Response: We appreciate the Reviewer’s comment on this point.
We and our colleagues revealed that 10nM of bradykinin (BK) could increase intracellular calcium concentration via ER calcium release, followed by SOC-mediated extracellular calcium influx in PAECs. (Takahashi R et al., Jpn Circ J 1997, 61, 1030-1036., Kimura M et al., J Hypertens 2000,18, 287-292., and Takeuchi K et al, Cardiovascular Research 2004, 62, 194– 201.). Thus, the effect of BK (10 nM) on intracellular Ca2+ response in PAECs was established. Although, we did not assess the effect of 1µM of BK on endothelial Ca2+ response in current study, Ana Rita Pinheiro et al., reported that BK (0.001-100 μM) caused [Ca2+]i rise in dose-dependent manner (Cell Commun Signal. 2013, 11, 70). Furthermore, our colleagues also revealed that 1 µM of BK activated endothelial nitric oxide synthase, which requires Ca2+ entry for sustained activation (Jiazhang Wei et al., Circ J 2013, 77, 2823-2830). From these reports, we strongly considered that 1µM of BK also increase intracellular calcium concentration via SOCE.

In preliminary study, we evaluated the effect of 10 nM to 1 µM BK (10 nM, 100 nM and 1µM) on ROS production, both 10 nM and 100 nM of bradykinin did not increase the ROS production, whereas 1 µM of BK increased ROS (data not shown). Thus, we selected 1 µM of BK to stimulate the ROS production in current study.

As reviewer 2 pointed, we agreed that the information which demonstrate the 1 µM of BK can increase intracellular Ca2+ via SOCE is useful to interpret our findings, however, previous findings strongly indicate that 1 µM of BK increases [Ca2+]via SOCE as described above. Furthermore, we need to upload the revised manuscript within 10 days after receiving decision letter. The 10 days is too short to resolve your suggestion. Thus, we discussed about this issue in Discussion section.

We have added the following text in Discussion section (page 9, lines 202-208):

Indeed, we and our colleagues revealed that BK (10 nM) could increase [Ca2+]i via ER Ca2+ release, followed by SOC-mediated extracellular Ca2+ influx in PAECs [16,25,26]. In addition, Ana Rita Pinheiro et al., also reported that BK (0.001-100 μM) could increase [Ca2+]i in a dose-dependent manner [27]. Furthermore, our colleagues also revealed that 1µ M of BK activated endothelial nitric oxide synthase, which requires Ca2+ entry for sustained activation [28]. On the basis of these findings, we used 1 µM of BK to introduce the increase of [Ca2+]i"
We have added the reference and changed reference number (page 13, lines 384-395):

25.          Takeuchi, K.; Watanabe, H.; Tran, Q.K.; Ozeki, M.; Sumi, D.; Hayashi, T.; Iguchi, A.;  
Ignarro, L.J.; Ohashi, K.; Hayashi, H. Nitric oxide: Inhibitory effects on endothelial cell calcium signaling, prostaglandin I2 production and nitric oxide synthase expression. Cardiovasc Res 2004, 62, 194-201.

26.          Kimura, M.; Watanabe, H.; Takahashi, R.; Kosuge, K.; Umemura, K.; Hayashi, H.; Ohashi, K.; Ohno, R. Inhibitory effect of insulin on bradykinin-induced venodilation. J Hypertens 2000, 18, 287-292.

27.          Pinheiro, A.R.; Paramos-de-Carvalho, D.; Certal, M.; Costa, C.; Magalhaes-Cardoso, M.T.; Ferreirinha, F.; Costa, M.A.; Correia-de-Sa, P. Bradykinin-induced Ca2+ signaling in human subcutaneous fibroblasts involves atp release via hemichannels leading to P2Y12 receptors activation. Cell Commun Signal 2013, 11, 70.

28.          Wei, J.; Takeuchi, K.; Watanabe, H. Linoleic acid attenuates endothelium-derived relaxing factor production by suppressing camp-hydrolyzing phosphodiesterase activity. Circ J 2013, 77, 2823-2830.

Furthermore, we added the following text in Results section (page 2 line 68-69):

“First, we investigated the effect of bradykinin (BK), which is a widely used pharmacological agent to evoke the SOCE, on cytosolic ROS production in primary cultured PAECs loaded with the ROS indicator 6-carboxy-2’ 7’-dichlorodihydrofluorescein di-(acetate, di-acetoxymethyl ester) (C-DCDHF-DA, 5 µM). 

Comment 2: The study showed the role of NOX in BK-induced ROS generation using

genic NOX blockers, apocynin or VAS. The authors also demonstrated by western blotting expression of several NOX subtypes, NOX2, NOX4 and NOX5 in endothelial cells. It is highly desirable or interesting to explore which one of NOXs or all of them are engaged in BK-induced ROS production.

Response: We thank you very much for your motivated about our study. We could not detect which NOX subtypes were activated by BK stimulation. As described in introduction section. we focused on determining whether Ca2+ mobilization from the ER, but not SOCE, is required to regulate NOX-derived ROS via Ca2+/CaM-dependent pathways in current study. Remaining challenges will be considered for further studies.

We have added the following text in Conclusion section (page 11, lines 312-313):

Further studies will be needed to elucidate which one of NOXs or all of them are engaged in BK-induced ROS production.

Round  2

Reviewer 1 Report

This article was revised appropriately.

I recommend accept

Reviewer 2 Report

The authors have revised the manuscript by adding more supporting literature and also modified the discussion according to the comments from the reviewers. However, the authors should request 3 months to perform the experiments and provide evidence to confirm that BK at the concentration used can induce internal calcium release and SOC-mediated extracellular calcium influx. Such information is essential for interpreting the experiment data and consolidating the study. I am happy with the approach the authors revise the manuscript in response to the second major point I raised on the initial submission.

 Author Response

Response to Reviewer 2

Comments and Suggestions: The authors have revised the manuscript by adding more supporting literature and also modified the discussion according to the comments from the reviewers. However, the authors should request 3 months to perform the experiments and provide evidence to confirm that BK at the concentration used can induce internal calcium release and SOC-mediated extracellular calcium influx. Such information is essential for interpreting the experiment data and consolidating the study. I am happy with the approach he authors revise the manuscript in response to the second major point I raised on the initial submission.
Response:
We thank very much for the reviewer’s kindly comment. According to reviewer #2’s suggestion, Ryugo Sakurada (the 1st author of this manuscript) and Chiaki Kamiya, who is a colleague at our laboratory, conducted additional experiments required by reviewer#2. We used to assess the intracellular Ca2+ concentration by using fluorescent indicator fura-2/AM. As shown a figure 1A (see the revised manuscript), 1μM of bradykinin rapidly increased fluorescence ratio (F340/F380) of fura-2 which expressed intracellular Ca2+ concentration and followed by a sustained increase in modified Tyrode’s solution containing Ca2+ (1 mM.). In the absence of extracellular Ca2+, BK (1μM) caused small and transient increase of F340/F380. From these result (figure 1A), we confirmed that BK (1μM) could induce internal Ca2+ release (ER Ca2+ release) and SOC-mediated extracellular Ca2+ influx. Furthermore, we assessed the effect of BAPTA/AM on ER Ca2+ release (figure 1B). BAPTA/AM (100 (100(100µM , the concentration we used) abolished the BK (1μM)-induced increase of the internal Ca2+ release. We thought that this information also strongly supported the interpreting our findings.

We added Chiaki Kamiya in the authors list because she contributed intellectual input to the research and paper. The author list has changed as below (p1 line 6-7):
“Ryugo Sakurada1, Keiichi Odagiri1,*, Akio Hakamata1, Chiaki Kamiya1, Jiazhang Wei 1,2, and Hiroshi Watanabe1”
.
We added the following text in Result section (page 2, lines 64-80):
“2.1. Bradykinin increased cytosolic Ca2+ concentration
Bradykinin (BK) is a widely used pharmacological agent to evoke the SOCE. It causes a biphasic elevation of intracellular Ca2+ concentration, which consists of an initial rise caused by Ca2+ mobilization from the ER and a subsequent influx of Ca2+ from the extracellular space [16,17]. First, we conformed the effect of bradykinin (BK) (1 μM) on cytosolic Ca2+ increase in primary cultured porcine aortic endothelial cells (PAECs). For the measurement of cytosolic Ca2+, cells were loaded fura-2/AM (2.5 μM). Figure 1A shows the time courses of changes in the fluorescence ratio (F340/F380) of fura-2 which expressed changes in the intracellular Ca2+ concentration. In the presence of 1 mM extracellular Ca2+, 1 μM of BK rapidly increased fluorescence ratio of fura-2 from basal levels of 0.87 ± 0.01 to a maximum of 3.89 ± 0.07 at 90 seconds (p < 0.01), followed by a sustained increase (2.95 ± 0.01 at 360 seconds, p < 0.01 versus baseline). In the absence of extracellular Ca2+, BK caused only a small and transient increase (basal levels of 0.59 ± 0.004 to 1.21 ± 0.06 at 90 seconds, p < 0.01). Pretreatment with 1,2-Bis (2 aminophenoxy) ethane-N,N,N’,N-tetraacetic acid tetraacetoxymethyl ester (BAPTA/AM, 100 μM) in Ca2+-free modified Tyrode’s solution abolished BK-induced Ca2+ responses (basal levels of 0.63 ± 0.004 to 0.64 ± 0.08 at 90 seconds, p = ns ) (Figure 1B). These results indicated that 1 μM of BK introduced SOCE, and BAPTA/AM eliminated the effect of BK on ER Ca2+ mobilization.”
We added the Figure 1 and figure Legend (page 3, lines 81-86).
We changed in Discussion section (page 8 line 221-222):
“On the basis of these findings, we confirmed that 1 μM of BK evoked SOCE in PAECs and used 1 μM of BK to introduce the increase of [Ca2+]i.”
We added the following sentence in Materials and Methods section: (page 10, line 276-280)
4.2. Measurement of endothelial Ca2+ concentration and ROS production
Intracellular Ca2+ concentration was measured using fura-2/AM, a fluorescent Ca2+ indicator. Porcine aortic endothelial cells were incubated in modified Tyrode’s solution composed of (mM) 150 NaCl, 2.7 KCl, 1.2 KH2PO4, 1.2 MgSO4, 1.0 CaCl2, and 10N-2-hydroxyethylpiperazine-N-2-ethanesulfonic acid (pH 7.4 at 25°C) with fura-2/AM (2.5 μM) for 45 minutes.”
We changed Materials and Methods section (page 10, line 276-291):
“Before measurements of Ca2+ and ROS, cells were subsequently washed three times with
modified Tyrode’s solution to remove the dye. The absorption shift of fura-2 that occurs upon binding to intracellular Ca2+ is determined by scanning the excitation spectrum between 340 and 380 nm while monitoring the emission at 510 nm. C-DCDHF-DA was excited at 490 nm with a xenon lamp, and emission signals were collected through a 510-nm long-pass filter. Fluorescent images of fura-2/AM and C-DCDHF-DA were acquired and quantified every 30 seconds from individual cells with a fluorescence analyzer (Aqua-cosmos, Hamamatsu Photonics K.K) using an ultra-high sensitivity camera. Changes in the fluorescence ratio (F340/F380) of fura-2 were used to express changes in the intracellular Ca2+ concentration.”
In addition, we changed the paragraph numbers and figure numbers in Results section and figure legends. Furthermore, we corrected some description to improved our manuscript. These changes was highlighted in red characters and also shown in underline in the manuscript.